# Structural Characterization of Glycerophosphorylated and Succinylated Cyclic β-(1→2)-d-Glucan Produced by *Sinorhizobium mliloti* 1021

**DOI:** 10.3390/polym12092073

**Published:** 2020-09-12

**Authors:** Hyojeong Lee, Seonmok Kim, Yohan Kim, Seunho Jung

**Affiliations:** 1Department of Bioscience and Biotechnology, Microbial Carbohydrate Resource Bank (MCRB), Center for Biotechnology Research in UBITA (CBRU), Konkuk University, Seoul 05029, Korea; lhjeong199@naver.com (H.L.); gkdurk9999@naver.com (S.K.); shsks1@hanmail.net (Y.K.); 2Department of Systems Biotechnology & Institute for Ubiquitous Information Technology and Applications (UBITA), Center for Biotechnology Research in UBITA (CBRU), Konkuk University, Seoul 05029, Korea

**Keywords:** cyclic polysaccharide, anionic cyclic β-(1→2)-d-glucan, *Sinorhizobium meliloti 1021*, MALDI-TOF, NMR, succinyl substituents, phosphglycerol substituents

## Abstract

*Rhizobia* produces different types of surface polysaccharides. Among them, cyclic β-(1→2)-d-glucan is located in the periplasmic space of rhizobia and plays an important role in the adaptation of bacteria to osmotic adaptation. Cyclic β-(1→2)-d-glucan (CG), synthesized from *Sinorhiozbbium meliloti* 1021, has a neutral and anionic form. In the present study, we characterized the exact chemical structures of anionic CG after purification using size exclusion s (Bio-Gel P-6 and P-2) chromatography, and DEAE-Sephadex anion exchange chromatography. The exact structure of each isolated anionic CG was characterized using various analytical methods such as nuclear magnetic resonance (NMR), attenuated total reflection Fourier transform infrared (ATR-FTIR) spectroscopy and matrix associated laser desorption ionization-time of Flight (MALDI-TOF) mass spectrometry. The precise chemical structures of novel anionic CG molecules were elucidated by various NMR spectroscopic analyses, including ^1^H, ^13^C, ^31^P, and 2D HSQC NMR spectroscopy. As a result, we discovered that anionic CG molecules have either glycerophosphoryl or succinyl residues at C6 positions of a neutral CG. In addition, the results of MALDI-TOF mass spectrometric analysis confirmed that there are two types of patterns for anionic CG peaks, where one type of peak was the succinylated CG (SCG) and the other was glycerophospholated CG (GCG). In addition, it was revealed that each anionic CG has one to four substituents of the succinyl group of SCG and glycerophosphoryl group of GCG, respectively. Anionic CG could have potential as a cyclic polysaccharide for drug delivery systems and a chiral separator based on the complexation with basic target molecules.

## 1. Introduction

*Rhizobia* are Gram-negative soil bacteria, which make Bacteroides in plants during root infection for nitrogen fixation [1]. The nodule development requires synthesis of signal molecules such as Nod factors for derivation of nodule development. *Rhizobium* spp. produces eight different types of surface polysaccharides for nodule development such as extracellular polysaccharide (EPS), lipopolysaccharide (LPS), capsular polysaccharide (CPS), cyclic β-(1→2)-d-glucan (CG), K-antigen poly-saccharide (KPS), neutral polysaccharide (NP), gel-forming-polysaccharide (GPS) and cellulose fibrils during root nodulation [2]. The cyclic β-(1→2)-d-glucan from *Rhizobium* spp. exists at the cytoplasmic or inner membrane in free-living conditions, which assists rhizobium strains in adapting to a hypoosmotic environment [3].

Cyclic β-(1→2)-d-glucan (CG) is found in the members of *Rhizobiaceae*, which are fast growing soil bacteria. These molecules are a class of unbranched cyclic oligosaccharides composed of β-(1→2)-d-glucans varying in size from 17 to 26 as a major neutral or anionic form [4]. The CG located in the periplasmic space of the bacteria plays important roles on the osmotic adaption as well as on the successful root-nodulation of *Sinorhizobium* species at the initial stage of the nitrogen fixation [3,5,6]. Recently, many studies have been conducted on inclusion complexes that use neutral CG. Since CG is very soluble, many studies have been conducted to improve insoluble compounds or bioactivity through an inclusion complex with various compounds such as ergosterol, fluorosine, indomethacin, paclitaxel, and atrazine [7,8,9,10,11]. CG molecules are larger and more diverse than cyclodextrin (CD), and have a flexible and twisted ring, so we can expect a completely different binding complex from CD, which has a smaller size and a hard ring [9]. Cyclodextrin (CD) is a cyclic polysaccharide composed of 6 (α-), 7(β-) or 8(γ-) glucopyranose units with α-1,4 glycosidic linkage. Donut shaped CDs with narrow and broad boundaries are known to make inclusion complexes through non-covalent interactions with various guest molecules that fit into the hydrophobic inner cavity of CDs [12,13,14].

Anionic CG are also located in the periplasmic space and play a role in osmotic regulation when the root nodulation for *Sinorhizobium* species to nitrogen fixation [6]. Various anionic substituents attached to CG are known to exist such as phosphoglycerol in A. *tumefaciens*, succinic acid in *B. abortus*, and both methyl malonic acid and succinic acid in *R*. *radiobcter* [15,16,17,18,19]. In the case of *S. meliloti*, it has been reported that the membrane derived phosphoglycerol group forms anionic CG, and its function is related to osmotic pressure regulation and vascularity [5,17]. It has been suggested that the anionic substituents of the surrounding cytoplasmic glucan significantly affect the movement potential across the outer membrane due to the net negative charge. Therefore, it will make an important contribution to maintaining the osmolarity of the surrounding vagina to the low osmotic concentration of the growth medium [20]. However, the exact chemical structures of all the substituents and the molecular weight distributions of anionic CG produced by *S. meliloti* have not been reported.

In this study, we characterized the chemical structures of anionic CG isolated from *Sinorhizobium meliloti* 1021 using matrix-assisted laser desorption/ionization time-of-flight mass spectrometry (MALDI-TOF MS) and various analytic methods, including nuclear magnetic resonance (NMR) Spectroscopy, attenuated total reflection Fourier transform infrared (ATR-FTIR) Spectroscopy and field emission scanning electron microscopy (FE-SEM).

## 2. Materials and Methods

### 2.1. Bacterial Strain and Culture Condition

*Sinorhizobium meliloti* 1021 was supplied by the Microbial Carbohydrate Resource Bank (MCRB) at Konkuk University, Korea. The bacterial strain used, *Sinorhizobium meliloti* 1021, was grown at 150 rpm on a rotary shaker at 30 °C in GMS medium (G: l-glutamic acid, M: d-mannitol, S: salts) [21]. For large amount isolation of cellular glucans, 30 mL pre-culture was inoculated into 500 mL GMS medium.

### 2.2. Preparation of Neutral and Anionic Cyclic β-(1→2)-d-Glucan

The *Sinorhizobium meliloti* 1021 was grown in GMS medium at 30 °C for 2 days. Cells were obtained by centrifugation (8000 rpm for 15 min at 4 °C), washed once with a 0.9% NaCl solution, then hot ethanol extraction was applied. Cells were extracted with 30 mL of 75% (*v/v*) ethanol at 70 °C for 30 min. After centrifugation, the remaining culture supernatant was precipitated by adding seven volumes of ethanol [6]. After centrifugation, the precipitate was separated on a Bio-Gel P-6 column for desalting. The fractions were assayed for carbohydrates using the phenol-sulfuric acid method. The desalted sample was applied to a DEAE-sephadex column for separation of neutral and anionic CG. The fractions containing anionic cyclic β-(1→2)-d-glucans were desalted by a Bio-Gel P-2 column.

### 2.3. Nuclear Magnetic Resonance (NMR) Spectroscopy

NMR spectroscopy was performed using Avance III-500 (Bruker, German). It was prepared by dissolving anionic CG in D2O 99.96%. The spectra of ^1^H NMR and ^13^C NMR and ^31^P NMR and ^1^H-^13^C HSQC(Heteronuclear Single Quantum Coherence) NMR were measured (500 MHz for ^1^H, 125 MHz for ^13^C, 202 MHz for ^31^P). ^13^C NMR analysis was performed using a general proton-decoupled method, and ^31^P NMR analysis was performed using two methods, proton-decoupled and proton-coupled, to obtain structural information of the glycerophosphoryl substituent. The rest of the parameters were taken by referring to the studies by K. MILLER et al. [17] and M. Batley et al. [22].

### 2.4. Matrix Assisted Laser Desorption/Ionization Time-of-Flight (MALDI-TOF) Mass Spectrometry

MALDI-TOF MS was obtained using the Voyager DE-STR (Applied Biosystems, Foster City, CA, USA) to obtain the MS spectrum, and 2,5-dihydroxybenzoic acid (DHB; Sigma-Aldrich, St. Louis, MO, USA) was used as a matrix in positive ion mode.

### 2.5. Attenuated Total Reflection-Fourier Transform Infrared (ATR-FTIR) Spectroscopy

The attenuated total reflectance (ATR) accessory was attached to an FTIR spectrometer (Spectrum Two FT-IR, Perkin Elmer, Waltham, MA, USA) to obtain a FTIR spectrum of the sample in the range of 4000–600 cm^−1^ at a resolution of 1 cm^−1^.

### 2.6. Field Emission Scanning Electron Microscopy (FE-SEM)

The Field Emission Scanning Electron Microscopy (FE-SEM) images of anionic CG were observed using FE-SEM (Hitachi S-4700, Tokyo, Japan). Samples were prepared in powder form after lyophilization. Carbon tape was attached on the brass stud and the sample was fixed. The powder samples were coated on the surface of a platinum layer prior to the FE-SEM analysis.

## 3. Results

### 3.1. DEAE-Sephadex Chromatography of CG

When ethanol extracts of *Sinorhizobium meliloti* 1021 were chromatographed on Bio-Gel P-6 chromatography, one major peak of oligosaccharide material was detected (data not shown). After further fractionation of this extract on DEAE-Sephadex, these oligosaccharides were found to be anionic in character, with neutral oligosaccharides. The column size was 3 cm × 23.5 cm, which was eluted first with 150-mL Mops buffer with no KCl and then with a 450-mL linear gradient beginning with 0 mM and ending with 250 mM KCl in the same buffer. Fractions (5 mL) were collected and assayed for total carbohydrates using phenol-sulfuric acid. The one neutral fraction (N) and four anionic fractions (designated F1, F2, F3, and F4) were present (Figure 1). Figure 2 shows the approximate structure of anionic CG containing phosphoglycerol and succinyl groups. The fraction N indicates neutral form with no charge. Each of anionic fractions were subjected to various analyses performed as described below.

### 3.2. Nuclear Magnetic Resonance (NMR) Spectroscopy Analysis

We analyzed anionic fractions isolated by anion-exchange chromatography. The chromatogram displayed five different fractions, indicating one neutral (N) and four anionic (F1 to F4) fractions (Figure 1). Each of these anionic fractions were subjected to nuclear magnetic resonance (NMR) spectroscopy. The N fraction showed a ^1^H NMR spectrum similar to that reported previously (Figure 3a) [19]. The NMR data for the remaining fractions F2, F3 and F4 are in Appendix A. The F1 fraction showed the peaks at 2.71 and 2.54 ppm assigned to the H-k and H-l resonances of the succinyl residues of these molecules (Figure 3b). As shown in Figure 3b, each of the three peaks at 3.85, 3.77 and 3.64 ppm was assigned to the H-h, H-i and H-j resonances of the phosphoglycerol residues of these molecules, respectively. In addition, two doublet signals at 4.48 and 4.38 ppm indicated H-g resonances of the glucosyl residues connected by phosphoglycerol substituents [17,19]. Further ^13^C NMR analyses were conducted to determine the structures of substituents. As shown in Figure 3c, each of three peaks appearing at 73.54, 70.71 and 62.13 ppm was assigned to the C-i, C-h and C-j resonances of the phosphoglycerol residues of these molecules, respectively. In addition, two more peaks were detected at 31.54 and 30.35 ppm indicating the succinyl residues. There was one other peak at 65 ppm assigned to H-g resonances of the glucosyl residues connected by phosphoglycerol substituents. The presence of phosphorus was also confirmed by ^31^P NMR spectroscopy, and resonances were a singlet at 1.4 ppm. The proton-coupled signals are also shown to be a quintet consistent with coupling to four protons (Figure 3e). These characteristic peaks suggest that anionic CG was substituted with phosphoglycerol or succinyl residue substituents.

### 3.3. Heteronuclear Single Quantum Coherence Spectroscopy (HSQC) Spectra Analysis

The heteronuclear single quantum coherence spectroscopy (HSQC) experiment was further conducted (Figure 4 and Appendix A). We confirmed the correlation between ^13^C NMR and ^1^H NMR through HSQC experiments. The NMR data for the remaining fractions F2, F3 and F4 are in Appendix A. In the HSQC experiments, the H-2′(S) was correlated with C-2′(S) and the H-3′(S) was correlated with C-3′(S), which represented the presence of succinyl substituents. In Figure 4, cross-peaks can be seen among the phosphoglycerol residue protons at 3.85, 3.77 and 3.64 ppm, and the two H-g protons are shown at 4.48 and 4.38 ppm. These results suggest that the anionic CG has glycerophosphoryl or succinyl residues at the C6 position.

### 3.4. Attenuated Total Reflection (ATR)-Fourier Transform Infrared (FTIR) Spectra Analysis

Fourier transform infrared spectroscopy was used to characterize the structure of anionic CG fraction ATR-FTIR spectra of anionic CG and neutral CG, shown in Figure 5. The broad absorption peak at 3355 cm^−1^ indicates the stretching vibration of the hydroxyl group [11]. Neutral CG had an absorption peak at 1645 cm^−1^ that was an O–H bending vibration. Anionic CG including F1, F2, F3 and F4 peaks had shifted O-H bending vibration peaks since intramolecular hydrogen bonding with esters of the succinate and hydroxyl group from the sugar backbone. The succinyl groups overlapped with the O–H bending vibration peak. However, the presence of succinyl substituents could be confirmed through NMR spectroscopic analysis as shown by Figure 3b and Appendix A. The phosphates group (P=O peak) showed the strong bands at 1158, 1165, 1176 and 1182 cm^−1^, respectively. In the neutral CG spectrum, the phosphate peak did not appear (highlighted in the blue box in Figure 5).

### 3.5. Matrix Assisted Laser Desorption/Ionization Time-of-Flight (MALDI-TOF) Mass Spectrometric Analysis

We analyzed anionic CG from *Sinorhizobium meliloti* 1021, with each of the separated fractions by matrix-assisted laser desorption/ionization time-of-flight (MALDI-TOF) mass spectrometry in a positive-ion mode (Figure 6 and Table 1). In the case of linear glucans, all the glucose monomers are linked through β-1, 2-glycosidic bonds, and the glucose at the end becomes the reducing end. However, compared to linear glucans, as they form cyclic glucans (CG) the reducing end disappears, resulting in a molecular weight without a single water molecule [23,24]. Reflecting this, the *m/z* values of anionic CG were also determined in MALDI-TOF analysis. Furthermore, it was also confirmed that both phosphoglycerol (PG) and succinyl (SU) groups were substituents for anionic CG. The spectrum from the F1 fraction showed each series of three ion species (Figure 6b and Table 1). The first series was a potassium cationized anionic CG with only succinyl substituents, which had [M + SU + K]^+^ ion species *(m/z* 3541.2, 3703.1, 3865.1, 4027.0, 4190.0). These ions series showed a molecular weight series in which only succinyl groups were substituted and composed of 21 to 24 cyclic-β-(1→2)-D-glucans. The difference of molecular weight of each unit was about 162, indicating one glucosyl linkage. The second series had [M + PG + 3Na − 2H]^+^ ion species (*m/z* 3617.2, 3779.1, 3941.1, 4103.0, 4266.0). These ion series showed a molecular weight series in which only phosphoglycerol groups were substituted and composed of 21 to 24 cyclic-β-(1→2)-D- glucans. The third series had [M + PG + K + 2Na - 2H]^+^ ion species (*m/z* 3633.1, 3795.0, 3957.0, 4119.0, 4280.9). As in the second of these ion series, only the phosphoglycerol group was substituted to the CG. The spectra of the F2, F3 and F4 fractions showed a series of molecular ions parallel to the F1 fraction, but each ionic species was different (Figure 6c,d and Table 1). The first series from the F2 fraction had two phosphoglycerol substituents and [M + 2PG + 4Na - 3H]^+^ ion species (*m/z* 3469.0, 3634.1, 3795.2, 3957.7, 4119.8, 4280.9) and composed 19 to 24 cyclic-β-(1→2)-d-glucans (Figure 6c). The second series also had only two phosphoglycerols without succinyl substituents, which had [M + 2PG + K + 3Na - 3H]^+^ ion species (*m/z* 3486.0, 3648.8, 3809.8, 3972.6, 4134.5, 4297.0). The third series had two succinyl substituents without phosphoglycerol substituents, which had [M + 2SU + K + Na - H]^+^ ion species (*m/z* 3503.0, 3664.5, 3827.7, 3989.0, 4150.5, 4313.0), and composed 20 to 24 cyclic-β-(1→2)-d-glucans. The F3 fraction showed a spectrum pattern similar to that of the F2 fraction (Figure 6d and Table 1). It had three types of ion spectrum series—two types of ion species had three phosphoglycerol substituents and the other type of ion species had three succinyl substituents. The first series had three phosphoglycerol substituents, [M + 3PG + 5Na - 4H]^+^ ion species (*m/z* 3649.0, 3803.0, 3971.3, 4132.1, 4294.0, 4458.2) and composed 19 to 24 cyclic-β-(1→2)-d-glucans. The second series also had only three phosphoglycerols without succinyl substituents, which had [M + 3PG + K + 4Na - 4H]^+^ ion species (*m/z* 3663.1, 3825.7, 3987.2, 4148.4, 4313.1, 4474.1). The third series had three succinyl substituents without phosphoglycerol substituents, which had [M + 3SU + K + Na - H]^+^ ion species (*m/z* 3679.7, 3840.5, 4002.3, 4166.3, 4329.1, 4490.1), and composed 21 to 25 cyclic-β-(1→2)-d-glucans. The F4 fraction consisted of two series of ion species (Figure 6e and Table 1). The first series had four phosphoglycerol substituents and [M + 4PG + Na]^+^ ion species (*m/z* 3872.8, 4034.7, 4196.2, 4358.6, 4519.6) and composed 20 to 24 cyclic-β-(1→2)-d-glucans. The second series also had only four succinyl without phosphoglycerol substituents, which had [M + 4SU + K]^+^ ion species (*m/z* 3888.3, 4050.3, 4213.0, 4373.6, 4535.8). As a result of the MALDI-TOF spectral analysis, each of phosphoglycerol and succinyl substituents had a separate molecular weight distribution. Thus, we confirmed that the two substituents were not present in one molecule at the same time, but existed separately in each molecule.

### 3.6. Field Emission Scannintg Electron Microscopy (FE-SEM) Analysis 

The surface morphology of anionic CG was observed by FE-SEM, as shown in Figure 7a–d. The figure demonstrates the morphology of F1, F2, F3 and F4 fractions. The increase in the anionic property changed the surface and morphology of anionic CG depending on the anionic characteristics. All fractions of anionic CG took on a rough-plate shape (Figure 7). The F1 fraction showed a relatively smooth surface (Figure 7a) and the F2 fraction started to become a smooth and rough surface (Figure 7b). Thus, the F4 fraction formed a rougher and coarser surface than any other fractions (Figure 7d). The fact that the surface differed from fraction to fraction might indicate that different morphology characteristics could be induced by increasing the anionic property of CG.

## 4. Conclusions

There have been some reports that phosphoglycerol substituents of anionic CG produced in *Sinorhizobium meliloti 1021* were derived from cell membrane lipids and that this negative CG is involved in the regulation of osmotic pressure [5,17]. However, studies to determine the exact position or number of substituents, and how anionic substituents are attached to the CG structure, have not been clearly conducted. In this study, we were able to find exact chemical structures of succinyl or glycerophosphoryl substituents constituting anionic CG using MALDI-TOF mass spectrometry and NMR spectroscopy. It was also confirmed that the anionic substituent of CG exists only in two types of succinyl groups or glycerophosphoryl groups. In other words, it was found that anionic CG that has two different substituents simultaneously was not present. That anionic CG would have potential in the fields of biotechnological applications like drug delivery systems and chirotechnology based on inclusion complexation with various guest molecules to increase solubility and bioavailability like CDs [25,26].

## Figures and Tables

**Figure 1 polymers-12-02073-f001:**
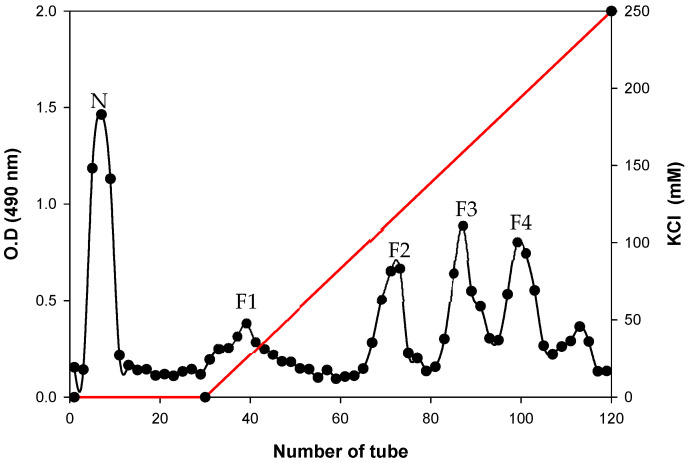
DEAE-Sephadex chromatography of neutral and anionic CG fractions from *Sinorhizobium meliloti* 1021.

**Figure 2 polymers-12-02073-f002:**
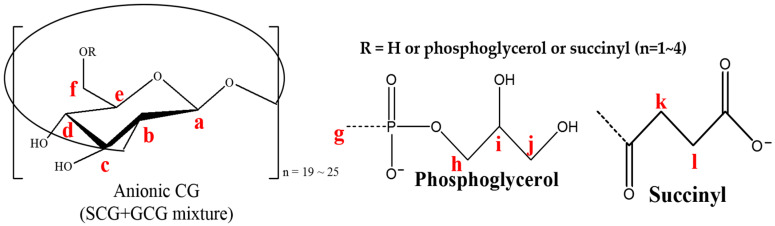
Structures of the anionic CG from Sinorhizobium meliloti 1021. Each of the two substituents, phosphoglycerol and succinyl, was attached to CG separately as SCG or GCG. No anionic CG containing both substituents was found.

**Figure 3 polymers-12-02073-f003:**
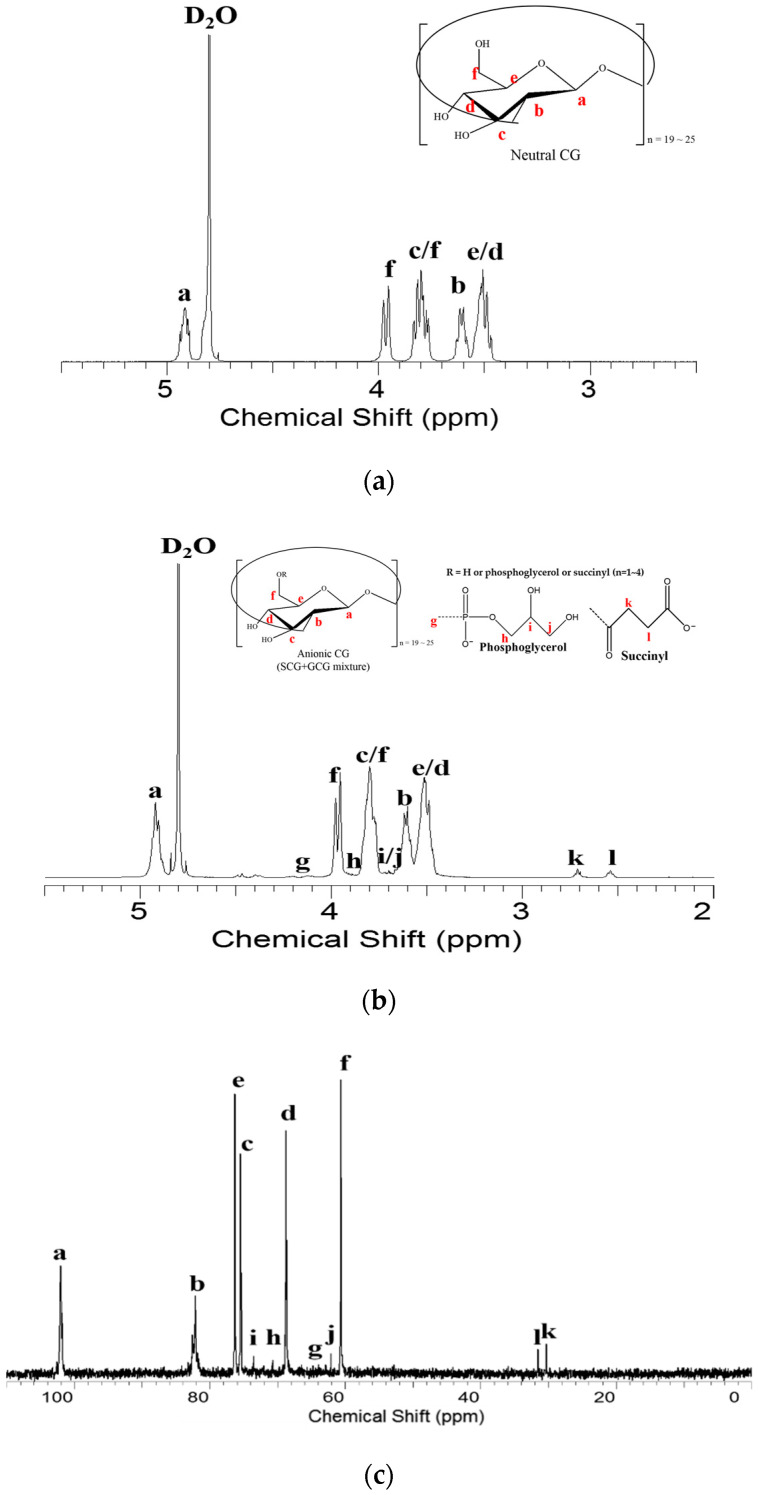
(**a**) ^1^H NMR spectra of N faction; (**b**) ^1^H NMR spectra of F1 faction; (**c**) ^13^C NMR spectra of F1 fraction; (**d**) The proton-decoupled ^31^P NMR spectrum of F1 fraction; (**e**) The proton-coupled ^31^P NMR spectrum of F1 fraction.

**Figure 4 polymers-12-02073-f004:**
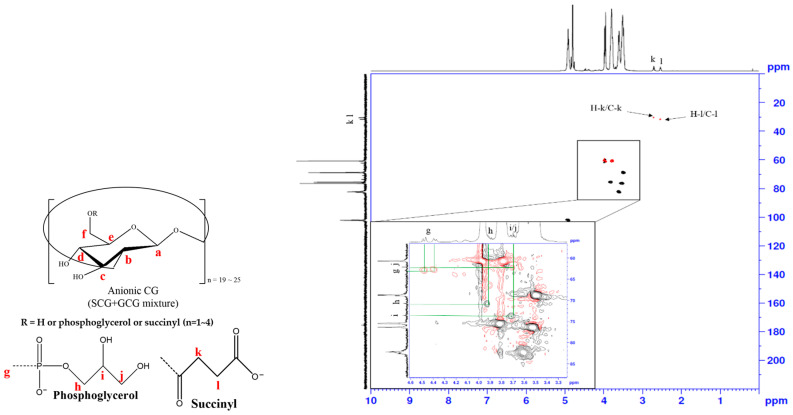
HSQC spectrum of F1 fraction.

**Figure 5 polymers-12-02073-f005:**
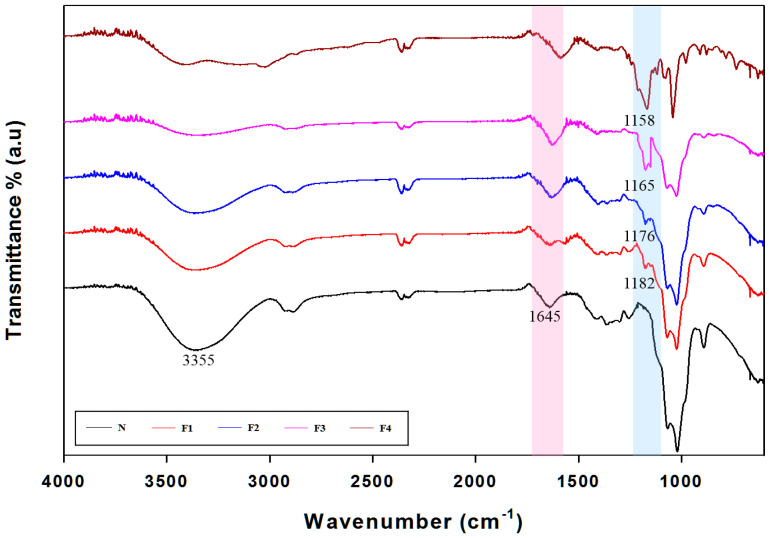
ATR-FTIR spectra of N fraction and anionic CG fraction region from 4000 to 600 cm^−1^.

**Figure 6 polymers-12-02073-f006:**
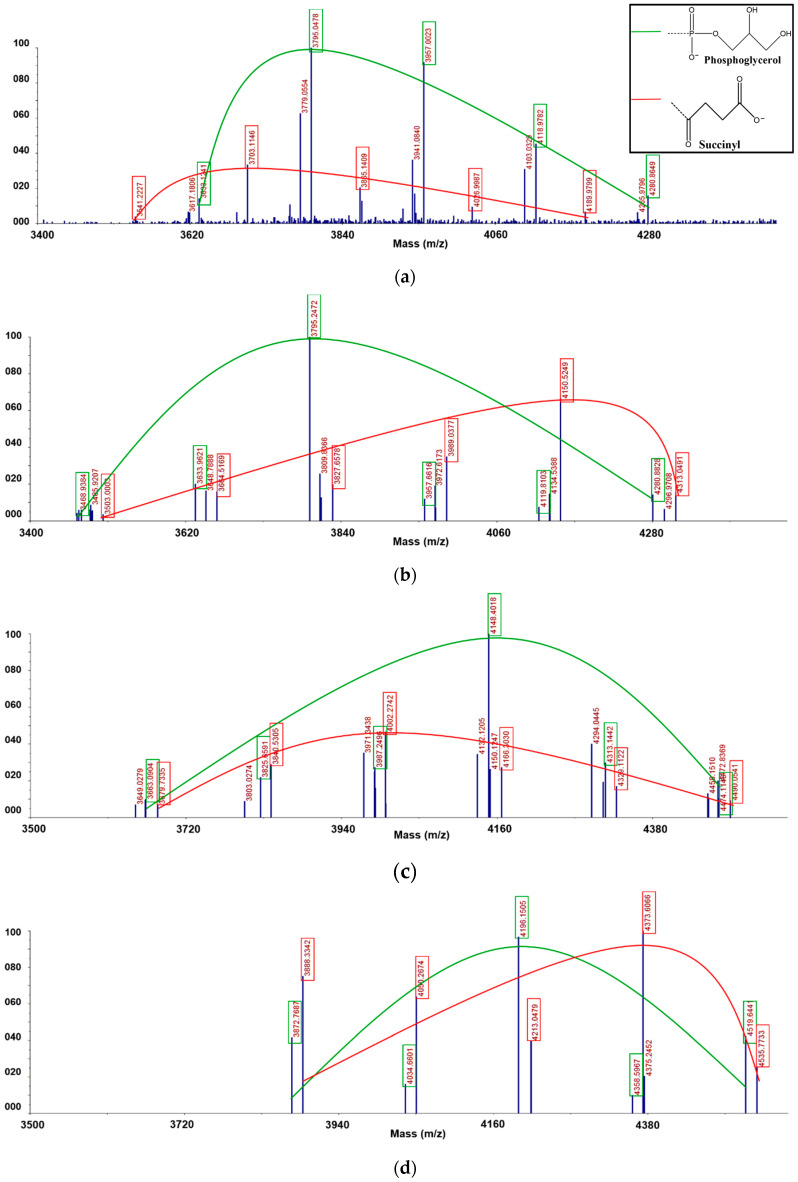
MALDI-TOF mass spectra of anionic CG obtained from (**a**) F1 fraction; (**b**) F2 fraction; (**c**) F3 fraction; (**d**) F4 fraction; acquired in positive ion reflector mode. The mass spectrums resulted from peak deisotoping based on the generic formula [C_6_H_12_O_6_]_n_ showing various sodium and potassium adduct peaks. Each green and red curve shows the molecular weight distribution of SCG and GCG, respectively.

**Figure 7 polymers-12-02073-f007:**
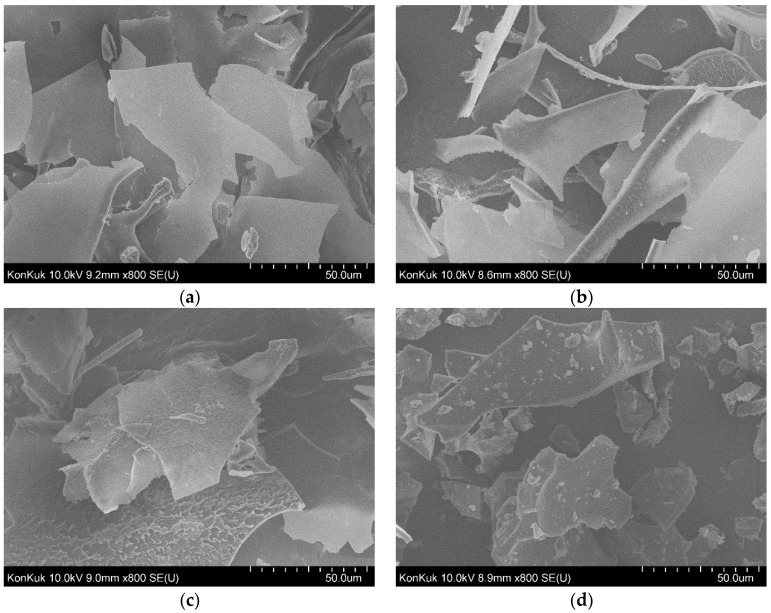
SEM images of (**a**) F1 fraction; (**b**) F2 fraction; (**c**) F3 fraction; (**d**) F4 fraction.

**Table 1 polymers-12-02073-t001:** MALDI-TOF MS data for F1 fraction, F2 fraction, F3 fraction and F4 fraction.

**Fraction**	**Measured Mass**	**Composition**	**Phosphoglycerol**	**Succinic Acid**
F1	3541.2227	Glc_21_+SU+K	0	1
	3617.1806	Glc_21_+PG+3Na-2H	1	0
	3633.1241	Glc_21_+PG+K+2Na-2H	1	0
	3703.1146	Glc_22_+Su+K	0	1
	3779.0554	Glc_22_ +PG+3Na-2H	1	0
	3795.0478	Glc_22_ +PG+K+2Na-2H	1	0
	3865.1409	Glc_23_ +SU+K	0	1
	3941.0840	Glc_23_+PG+3Na-2H	1	0
	3957.0023	Glc_23_+PG+K+2Na-2H	1	0
	4026.9987	Glc_24_+SU+K	0	1
	4103.0329	Glc_24_+PG+3Na-2H	1	0
	4118.9782	Glc_24_+PG+K+2Na-2H	1	0
	4189.9799	Glc_25_+SU+K	0	1
	4265.9796	Glc_25_+PG+3Na-2H	1	0
	4280.8649	Glc_25_+PG+K+2Na-2H	1	0
**Fraction**	**Measured Mass**	**Composition**	**Phosphoglycerol**	**Succinic Acid**
F2	3468.9384	Glc_19_+2PG+4Na-3H	2	0
	3485.9207	Glc_19_+2PG+K+3Na-3H	2	0
	3503.0003	Glc_20_+2SU+K+Na-H	0	2
	3633.9621	Glc_20_+2PG+4Na-3H	2	0
	3648.7888	Glc_20_+2PG+ K+3Na-3H	2	0
	3664.5169	Glc_21_+2SU+K+Na-H	0	2
	3795.2472	Glc_20_+2PG+4Na-3H	2	0
	3809.8366	Glc_20_+2PG+ K+3Na-3H	2	0
	3827.6578	Glc_21_+2SU+K+Na-H	0	2
	3957.6616	Glc_21_+2PG+4Na-3H	2	0
	3972.6173	Glc_21_+2PG+K+3Na-3H	2	0
	3989.0377	Glc_22_+2SU+K+Na-H	0	2
	4119.8103	Glc_22_+2PG+4Na-3H	2	0
	4134.5388	Glc_22_+2PG+K+3Na-3H	2	0
	4150.5249	Glc_23_+2SU+K+Na-H	0	2
	4280.8828	Glc_23_ +2PG+4Na-3H	2	0
	4296.9708	Glc_23_+2PG+K+3Na-3H	2	0
	4313.0491	Glc_24_+2SU+K+Na-H	0	2
**Fraction**	**Measured Mass**	**Composition**	**Phosphoglycerol**	**Succinic Acid**
F3	3649.0279	Glc_19_+3PG+5Na-4H	3	0
	3663.0904	Glc_19_+3PG+K+4Na-4H	3	0
	3679.7335	Glc_20_+3SU+3K+Na-3H	0	3
	3803.0274	Glc_20_+3PG+5Na-4H	3	0
	3825.6591	Glc_20_+3PG+K+4Na-4H	3	0
	3840.5305	Glc_21_+3SU+3K+Na-3H	0	3
	3971.3438	Glc_21_+3PG+5Na-4H	3	0
	3987.2496	Glc_21_+3PG+K+4Na-4H	3	0
	4002.2742	Glc_22_+3SU+3K+Na-3H	0	3
	4132.1205	Glc_22_+3PG+5Na-4H	3	0
	4148.4018	Glc_22_+3PG+K+4Na-4H	3	0
	4166.3030	Glc_23_+3SU+3K+Na-3H	0	3
	4294.0445	Glc_23_+3PG+5Na-4H	3	0
	4313.1442	Glc_23_+3PG+K+4Na-4H	3	0
	4329.1122	Glc_24_+3SU+3K+Na-3H	0	3
	4458.1510	Glc_24_+3PG+5Na-4H	3	0
	4474.1148	Glc_24_+3PG+K+4Na-4H	3	0
	4490.0541	Glc_25_+3SU+3K+Na-3H	0	3
**Fraction**	**Measured Mass**	**Composition**	**Phosphoglycerol**	**Succinic Acid**
F4	3872.7687	Glc_20_+4PG+Na	4	0
	3888.3342	Glc_20_+4SU+K	0	4
	4034.6601	Glc_21_+4PG+Na	4	0
	4050.2674	Glc_21_+4SU+K	0	4
	4196.1505	Glc_22_+4PG+Na	4	0
	4213.0479	Glc_22_+4SU+K	0	4
	4358.5967	Glc_23_+4PG+Na	4	0
	4373.6066	Glc_23_+4SU+K	0	4
	4519.6441	Glc_24_+4PG+Na	4	0
	4535.7733	Glc_24_+4SU+K	0	4

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
