# Peer review of "Structural Characterization of Glycerophosphorylated and Succinylated Cyclic β-(1→2)-d-Glucan Produced by *Sinorhizobium mliloti* 1021"

_polymers, 2020, doi:10.3390/polym12092073_

Round 1
Reviewer 1 Report
This is an interesting article. Few corrections should be carried out prior to acceptance:
-Quality of figures 2 and 3 should be improved
-Also the font size in figure 6 is not clear
-Some sentences should be rephrased.
Reviewer 2 Report
The paper reports the extraction by chromatography of several fraction of cyclic glucans from Sinorhizobium meliloti 1021 followed by their characterizations (NMR, MALDI-TOF, IR, SEM). The characterization proves that extracted cyclic glucans are substituted either by succinyl units or by glycerol-phosphate units. The characterizations are convincing. The collected data on the structure of these glycans are interesting for biologists and for the preparation of inclusion complexes (in a similar way to the use of common cyclodextrins). My remarks are minor, and I thus recommend this paper for publication
Minor remarks:
- Regarding the 31P NMR analysis, the authors should indicate in the description of the methods the sequence implemented (4 possibilities depending upon irradiation is applied or not during the delay and/or during the acquisition (POWERGATE, INVERSE GATE ???). It is an important information to let the reader know whether the spectrum is quantitative (is there a NOE effect) and decoupled or nor with the protons). I found finally the information because both spectra are shown in supplementary materials. When 31P is used and reported in the paper, it is important to mention systematically which spectrum is reported (decoupled or not). The same question is valid for 13C even though I suppose that the authors achieved irradiation both during the delay (presence of NOE effect) and the acquisition (decoupling with the protons). The delay times between the pulses should also be indicated.
- In the description of the 1H NMR, protons are labelled with numbers. It can be confusing for the reader because it is difficult to know if the numbers shown on the spectrum refers to the label of the carbon of to the number of protons for the signals. The use of letters rather than numbers for the labelling of protons avoid any confusion.
- The quality of the NMR spectra is poor. The numbers are small and difficult to read. This can be improved with common softwares (topspin and mestrenova).
- The authors commented clearly on the use of MALDI-TOF to highlight the nature of the substituent of cyclic glucans. I assume that MALDI-TOF is also useful to prove the cyclic nature of the glucans, but I found no comment regarding this point by reading the text. This could be a known story, but a small reminder could be useful.
